# Region-specific cardiovascular risk prediction from non-contrast chest computed tomography

**Vineet K Raghu**[1] iD                                   VRAGHU@MGH.HARVARD.EDU
**Daniel W Oo**[1]                                           DAOO@MGH.HARVARD.EDU
**Leonard Nurnberg** [2,3]                           LNURNBERG@BWH.HARVARD.EDU
**Audra Sturniolo**[1]                              ASTURNIOLO@MGH.HARVARD.EDU
**Douglas P Kiel**[4]                                         KIEL@HSL.HARVARD.EDU
**Hugo JWL Aerts**[2,3]                                 HAERTS@BWH.HARVARD.EDU
**Pradeep Natarajan**[5]                          PNATARAJAN@MGH.HARVARD.EDU
**Michael T Lu**[1]                                             MLU@MGH.HARVARD.EDU

[1] *Cardiovascular Imaging Research Center, Massachusetts General Hospital*

[2] *Program for Artificial Intelligence in Medicine, Brigham and Women's Hospital*

[3] *Radiology and Nuclear Medicine, CARIM and GROW, Maastricht University*

[4] *Hinda and Marcus Arthur Institute for Aging Research, Beth Israel Deaconess Hospital*

[5] *Department of Medicine, Massachusetts General Hospital*

**Editors:** Accepted for publication at MIDL 2025

## Abstract

Accurate cardiovascular risk scores can help direct preventive treatment to those who would maximally benefit. Current scores rely on established risk factors, but imaging may contain additional information to find high-risk patients. Here, we developed and tested a system called CT-CV-Risk to estimate cardiovascular risk from non-contrast chest CT images. We find that CT-CV-Risk predicts risk complementary to established clinical risk scores.

**Keywords:** CT imaging, cardiovascular risk, convolutional networks, segmentation

## 1. Introduction

Cardiovascular (CV) risk calculators are the basis for clinician-patient discussions to inform prevention strategies (Arnett et al., 2019). Clinical standard calculators (e.g., Pooled Cohort Equations or PCE and PREVENT) rely on established risk factors (e.g., age, cholesterol), but additional high-risk individuals could be identified using novel scores (Goff et al., 2014; Khan et al., 2024). For example, the coronary calcium score (CAC) measures calcified plaque on computed tomography (CT) scans. CAC is recommended to identify high-risk individuals (Greenland et al., 2018). CT may contain information beyond risk factors and CAC to inform CV risk; however, CTs are high-dimensional, leading to model overfitting.

Here, we develop CNNs applied to non-contrast chest CTs to estimate cardiovascular risk (we call CT-CV-Risk). We use TotalSegmentator (Wasserthal et al., 2023) to simplify CTs into regions of interest (ROIs; heart, aorta, right and left lung), and train separate networks on each ROI. Prior approaches automatically measure CAC (Zeleznik et al., 2021) and predict risk using a heart ROI (Chao et al., 2021; van Velzen et al., 2019). The proposed CT-CV-Risk approach is novel in that it uses the publicly available TotalSegmentator to define ROIs, incorporates multiple ROIs, and is tested in external validation.

## 2. Methods

**Datasets** We trained models on 60% of the National Lung Screening Trial's (NLST) (Team, 2011) chest CT arm (24,930 scans from 15,262 participants), and tested on a held-out 40% who had no history of heart attack or stroke (9,930 participants). We trained CT-CV-Risk to predict 10-year CV mortality risk based on ICD-10 codes for any CV-related death. We externally tested our models on Brigham and Women's hospital patients (N=6650), aged 40-75 who had a non-contrast, non-ECG gated chest CT in 2013-2018 and no history of heart attack, stroke, or coronary revascularization. The primary outcome for external testing was 10-year heart attack, ischemic stroke, or coronary revascularization identified by ICD-10 and CPT codes and confirmed via manual review of patient charts.

 **CT Preprocessing and Model Training** An overview of the CT analysis procedure is given in Figure 1. For each CT, we cropped the full volume to bounding boxes around each ROI and masked voxels outside the ROI. We then padded each ROI to a square region based on the longest axis length and resized to 96x96x32 for the lungs and aorta and 64x64x32 for the heart. We used a 3D Densenet169 architecture for each CNN. We trained each model for 80 epochs using an ADAM optimizer with a learning rate of $1e-4$ for 40 epochs and $1e-5$ for 40 epochs. We used random data augmentation including up to 5 voxels of translation, $\pi/10$ radians of rotation, and 10% zoom in/out. During inference, we used test-time augmentation for six iterations, including rotation, zoom, and brightness/contrast adjustments. The final CT-CV-Risk model was a logistic regression using the individual predictions from each CNN model as input to output 10-year CV mortality risk.

## 3. Results

**Internal Testing** We summarized results in Table 1. Models trained to estimate risk using the heart ROI had the highest discrimination, but there is added information in the aorta and lungs to assess risk. The composite CT model (CT-CV-Risk) had better discrimination than the CAC score (0.71 vs 0.66) and was complementary to a composite score of established CV risk factors (0.74 vs. 0.71). Further, we found that the CT-CV-Risk score had better calibration than the CAC score or established risk factors.

 **External Testing** In external testing, we found that the CT-CV-Risk model had higher discrimination than clinical standard risk scores; however, CT-CV-Risk underestimated risk since the outcome included non-fatal events.

## 4. Conclusions

- ROI segmentation and CNNs can predict CV risk in external testing
- Multiple ROIs beyond the heart carry unique information to predict risk
- A CT-based approach may improve CV risk prediction beyond established scores

## Acknowledgments

We thank the National Cancer Institute for use of the National Lung Screening Trial datasets. This work was supported by NHLBI K01HL168231 and AHA award 935176.

Table 1: Discrimination and Calibration - Internal Testing (N=9,930)

|  | C-statistic (95% CI) | Observed-Expected Ratio |
|---|---|---|
| CT-CV-Risk | 0.71 (0.68,0.74) | 1.00 |
| Conventional Risk Factors + CT Findings | 0.71 (0.68,0.74) | 1.12 |
| CT-CV-Risk + Conventional Risk Factors + CT Findings | 0.74 (0.70,0.77) | 0.98 |
| Heart ROI Only | 0.69 (0.66,0.72) | 0.96 |
| Aorta ROI Only | 0.67 (0.63,0.70) | 1.08 |
| Lungs ROIs Only | 0.63 (0.60,0.66) | 1.00 |
| CAC Score | 0.66 (0.63,0.70) | 1.15 |

Table 2: Discrimination and Calibration - External Testing (N=6,650)

|  | C-statistic (95% CI) | Observed-Expected Ratio |
|---|---|---|
| CT-CV-Risk | 0.73 (0.71,0.76) | 3.60 |
| PCE Risk Score | 0.64 (0.61,0.67) | 1.02 |
| PREVENT Risk Score | 0.64 (0.61,0.67) | 1.65 |

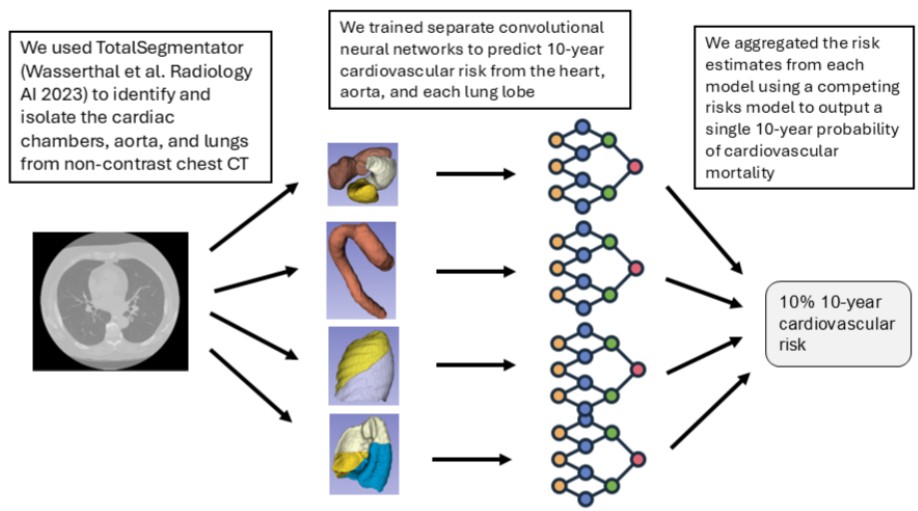

Figure 1: CT-CV-Risk Schematic

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
