# OpenReview forum: "Region-specific cardiovascular risk prediction from non-contrast chest computed tomography"
_MIDL.io/2025/Short_Papers — MIDL 2025 - Short Papers_

### Official Review · Reviewer_8fje · 2025-04-24

**Rating:** 4
**Confidence:** 5

**Summary:**

The authors propose to use a CNN to classify patients into low/high CVD risk based on cropped chest CT regions-of-interest (ROIs). These ROIs are obtained with a TotalSegmentator model. Authors find that a model trained on heart ROIs provides the best risk assessment, but the lungs and aorta also contain relevant information. Moreover, this information is complementary, as a model combining heart, lungs, and aorta performs best.

**Strengths:**

- The paper is clear, and the figures are informative.
- Creative use of off-the-shelf models to answer a clinical question.
- It's an interesting finding that the aorta and lung ROIs also contain relevant info for CVD risk.
- The authors use an external test set to validate their model.

**Weaknesses:**

- The paper does not describe how the 'composite CT model' was defined.
- Direct CT-->CVD risk modelling from chest CT with a CNN is not necessarily novel (see, e.g., https://doi.org/10.1117/12.2512400).

---

### Decision · Program_Chairs · 2025-05-01

Accept